# Perceptions of Antenatal Exercise in Pregnant Females and the Impact of COVID-19

**DOI:** 10.3390/ijerph191710635

**Published:** 2022-08-26

**Authors:** Madeleine France-Ratcliffe, Nicola D. Hopkins, David A. Low, Matthew S. Cocks, Helen Jones, Kayleigh S. Sheen, Victoria S. Sprung

**Affiliations:** 1Research Institute for Sport & Exercise Sciences, Liverpool John Moores University, Tom Reilly Building, Byrom Street, Liverpool L3 3AF, UK; 2Faculty of Health, School of Psychology, Liverpool John Moores University, Tom Reilly Building, Byrom Street, Liverpool L3 3AF, UK

**Keywords:** pregnancy, exercise, COVID-19, physical activity, interviews

## Abstract

Exercise during pregnancy presents many benefits for the mother and baby. Yet, pregnancy is characterised by a decrease in exercise. Studies have reported barriers to antenatal exercise. The coronavirus (COVID-19) pandemic may have further exacerbated barriers to antenatal exercise as pregnant females faced many challenges. Rich, in-depth exploration into pregnant female’s perceived barriers to antenatal exercise during COVID-19 is imperative. Questionnaires reporting physical activity levels were completed by all participants (*n* = 14). Semi-structured interviews were conducted between November 2020 and May 2021 in the UK. Interviews were analysed using thematic analysis and revealed four main themes: ‘Perceptions of being an active person shaping activity levels in pregnancy’, ‘How do I know what is right? Uncertainty, seeking validation and feeling informed’, ‘Motivators to antenatal exercise’ and ‘A process of adaptations and adjustment’. Findings indicate that the COVID-19 pandemic exacerbated barriers to antenatal exercise and highlight the importance of direct psychosocial support and clear, trustworthy information. Findings also support the fundamental need for better education amongst healthcare professionals regarding antenatal exercise.

## 1. Introduction

Current guidelines in the United Kingdom (UK) recommend that pregnant females, without contraindication, perform at least 150 min of moderate-intensity exercise per week [1]. Meeting these guidelines has benefits for both the mother and foetus, for example being at a decreased risk of gestational diabetes mellitus, pre-eclampsia and fewer foetal complications [2]. Exercise during pregnancy also decreases the odds of developing antenatal depression by 67% [3]. Yet, despite these well-documented benefits of exercise, pregnancy is often characterised by a substantial decrease in exercise levels [4,5]. Several studies have explored pregnant female’s experiences of, and barriers to, antenatal physical activity (PA). Barriers reported include insufficient time, lack of childcare, lack of facilities, discomfort, fatigue, and concerns about risk of exercise during pregnancy [6,7,8,9,10,11,12,13,14].

The recent COVID-19 pandemic may have exacerbated barriers to antenatal exercise. Due to pregnancy being an immune-compromised state, pregnant females could be more vulnerable to viral respiratory infections, including COVID-19 [15]. The COVID-19 outbreak presented many changes for the general population in the UK, including social distancing rules, and various local and nationwide ‘lockdowns’ (stay-at-home order) imposed by the UK Government. Pregnant females, specifically, faced additional significant restrictions. For instance, females were advised to ‘shield’ (stay-at-home) during their third trimester and have limited in-person appointments [16]. A large online survey of 900 pregnant and recently postpartum females in Canada found that 64% of participants reported reduced PA with the onset of isolation measures [17]. Yet, females with increased PA levels had significantly lower scores for anxiety and depression during the COVID-19 pandemic restrictions [17]. This highlights the importance of PA and exercise in pregnancy, and the need to understand pregnant female’s approaches and attitudes to PA and exercise during the pandemic to design optimal interventions against a background of social restrictions.

There is an absence of rich, in-depth qualitative data to reflect pregnant female’s views and perceptions of antenatal exercise during COVID-19. Therefore, the aim of this study was to understand perceived barriers to antenatal exercise during the COVID-19 pandemic.

## 2. Materials and Methods

### 2.1. Ethical Approval

The study conformed to the Declaration of Helsinki and was approved by the National Health Service Liverpool Central Research Ethics Committee (Project ID: 263600). All participants were informed of the protocol verbally and in writing before providing written informed consent prior to participation.

### 2.2. Design

This qualitative study, using individual semi-structured interviews, was part of a larger observational trial examining the impact of PA on glucose control throughout pregnancy (unpublished study). Participants firstly conducted home-based health assessments (e.g., capillary blood sampling, blood pressure, continuous glucose monitoring), after which they completed a survey to collect demographic and PA information. Participants were instructed to continue their usual daily activities throughout monitoring periods and no intervention or treatment took place. Semi-structured interviews with a subset of participants from the larger observational trial, who indicated willingness to take part, were organised according to participant availability within the following month after survey was completed. Purposive sampling was employed by asking participants who were in various stages of gestation whether they would like to take part in the semi-structured interviews (i.e., if there were many participants in trimester 2 who had been interviewed, no more participants in trimester 2 were asked) to increase diversity/representation and enable sharing of broader experiences across the sample [18].

### 2.3. Participants

Participants were required to be a resident of Great Britain, pregnant with a single child, aged 18–45 years, a non-smoker for at least 6 months, not pregnant via in vitro fertilisation and not taking anti-inflammatory medications. These eligibility requirements were deemed appropriate as part of the aforementioned larger observational trial to enable assessment of relationship between PA and glucose control. There was no restriction for participation based on gestational age. Participants were recruited via social media advertisements. Ultimately, participation was voluntary and all those who volunteered to take part in the interview during the study (*n* = 14) were included.

### 2.4. Data Collection

Participant characteristics: An online survey was used to record the following demographic information: age, medical conditions, education level, employment status, occupation.

Physical activity: The short-form International Physical Activity Questionnaire (IPAQ) [19] was administered to all participants online. Participants were asked how many days they walked and did moderate and vigorous PA for at least 10 min at a time within the last 7 days. PA using the IPAQ was classified as ‘high’, ‘moderate’, or ‘low’. Definitions of these classifications are reported by the IPAQ Research Committee (2005) [20].

Interviews: Fourteen semi-structured, one-to-one interviews of up to 45 min (median = 27 min) in duration were conducted between November 2020 and May 2021 by MF. During this period, several COVID-19 restrictions came into effect in the UK. These restrictions included two separate national lockdowns as well as local lockdowns [21], all restrictions were rapidly changing and may have an important effect on outcomes. Females participated in the interviews via telephone or video call (Zoom), according to their preference. Before participants were interviewed, they met with the researcher at least twice previously as part of the larger observational trial. Data collection occurred over a 7-month period, which enabled the capture of different experiences with regard to varying government rules and restrictions dictated by government in response to the COVID-19 pandemic. Prior to the researcher beginning the interview, participants were asked if they had any remaining questions about the study which were addressed accordingly. Upon completion of the interview schedule, the participant was asked if they had anything else to add before the interview was formally concluded. All interviews were digitally recorded, transcribed verbatim and transferred into NVivo (Release 1.5) for analysis. Transcripts were not returned to participants for comment.

### 2.5. Interview Schedule

An interview schedule was used to direct conversation regarding experiences of exercise prior to pregnancy, perceptions of exercise during pregnancy, and experiences of exercise during pregnancy throughout the COVID-19 pandemic. The schedule was devised from the research objective and consisted of open-ended questions. Probes were utilised where appropriate (see Appendix A).

### 2.6. Analysis

Transcripts were analysed using NVivo (Release 1.5, QSR International, Doncaster, Australia). Reflexive thematic analysis was used to identify parallels and similarities in participants’ views on antenatal exercise [22,23]. The thematic analysis method follows a systematic, six-step approach whereby the researcher firstly familiarises themselves with the data, generates initial codes, searches for themes, reviews these themes, defines and names themes, and finally produces the report [22,23]. Disconfirmatory evidence, where identified, is highlighted within the text. MF and KS were involved in all stages of the thematic analysis. This study is rooted in relative epistemology (i.e., reality is varied and multiple) and adopts a social constructionist view of reality whereby perspectives acknowledge that multiple realities are subjective and socially constructed by cultural, social, ethnic and gender factors [24]. The study findings are shaped by communications obtained and rely on interpreting perceptions of interviewed participants.

### 2.7. Reflexivity

The first author, MF, was a female PhD researcher and had no personal experience of pregnancy though shared the experience of the national COVID-19 restrictions (variation related to the local [County] lockdown tier system. Clear focus throughout the interview process on exploring pregnant female’s perspectives was maintained to reduce bias. Reflexivity was facilitated by ongoing reflection throughout the data collection and analysis process. The first author also considered their position as having no first-hand experience of pregnancy as an advantage during data collection and analysis. This position potentially empowered participants to share their unique experiences, without being influenced by any suggestion or preconceived ideas from the researcher. KS, who has extensive previous experience in thematic analysis, and knowledge of the field enhanced the depth of interpretation of data. KS is a female Psychology Lecturer with no personal experience of childbearing. Despite involvement in study design and write-up, the roles of the remaining co-authors did not directly influence interpretation of data.

## 3. Results

### 3.1. Participant Demographics

Participants were aged between 30 and 38 years (34 ± 2 years), and week of gestation ranged from 13 to 37 weeks (26 ± 7 weeks). Self-reported pre-pregnancy body mass index (BMI) of participants was 25 ± 4 kg/m^2^. All participants were in a relationship or married and living together. Half of the participants had previously given birth to a child (*n* = 6 given birth once previously, *n* = 1 given birth twice previously). Two participants disclosed pre-existing medical conditions prior to pregnancy (*n* = 1 hypothyroidism, *n* = 1 essential thrombocytosis), one participant reported a medical condition obtained during pregnancy (hiatus hernia), whilst all other participants reported no medical conditions. See Table 1 for demographic information.

### 3.2. Thematic Analysis

Thematic analysis generated four main themes, each with sub-themes (Table 2).

#### 3.2.1. Theme 1: Perceptions of Being an ‘Active Person’ Shaping Activity Levels in Pregnancy

This theme reflects participants’ views on how their attitude towards, and level of activity during pregnancy, were often framed in the context of how active a person they viewed themselves. This included general views of being an active person (or not so active) in their daily lives and highlighted how, often, these perceptions of being an active person shaped their current activity levels. Within this, participants often drew upon their previous experiences of exercising prior to and during pregnancy, and how this shaped their perspectives during their current pregnancy.

i.Previous activity habits shaping current habits

In this subtheme participants reflected on their previous activity, and how this shaped both their current activity levels and their views on how active they felt they could be during their current pregnancy.


*“So I was always very active as a child and a young person, so it was more kind of carrying on what was normal for me.”*
(Participant 11)

Participants who were active previously reported continuing what was deemed ‘normal’ activity for them throughout pregnancy, whilst others who classed themselves as previously inactive also reported continuing what was deemed ‘normal’ activity for them.


*“I’ll never class myself as someone who was super fit or active”.*
(Participant 4)

ii.Seeing the value in being active: the perceived importance of exercise

This subtheme reflects participants’ views on the importance of exercise often shaped by their previous levels of activity and recognition of associated benefits, such as feeling physically better, switching off from daily stress and enjoying the challenges of exercise. Some participants felt that it was important to them to continue their activity levels to maintain their identity of being an ‘active person’, although noted that some activities needed to be adapted during pregnancy.


*“I think you’ve got to, because I think if you do exercise anyway, then you can’t just stop, and just go crazy.”*
(Participant 7)

A number of participants felt they needed to be an advocate for antenatal exercise. Some participants felt that this was setting a good example for their children, whilst others felt that sharing their experiences could help other pregnant females.


*“We will go out for a family run and my partner takes the toddler in the buggy. So we all run together and get something out of it, which is nice. And I suppose it’s setting an example that actually and showing [her children] that it’s okay to do exercise when you’re pregnant, because it can be quite a taboo subject in a way.”*
(Participant 14)

#### 3.2.2. Theme 2: How Do I Know What Is Right? Uncertainty, Seeking Validation and Feeling Informed

Participants talked about the importance of information to support their choices regarding antenatal exercise. Participants expressed feelings of uncertainty about what exercises (if any) they should or should not perform during pregnancy, and their pursuit to find credible sources to inform and validate their activity.

i.Uncertainty and seeking validation

In this subtheme, participants felt uncertain about the ‘rights and wrongs’ of antenatal exercise and as a result were seeking validation for their activity choices to provide confidence and peace of mind. Participants turned to internet search engines (e.g., Google), speaking to others, observing what other pregnant females do and relevant books.


*“I was running a little bit, and I feel like, I don’t know if you should be or you shouldn’t be, and it’s one of those blurry lines.”*
(Participant 5)

Many participants perceived the advice received (or lack thereof) regarding exercise from health care professionals (HCPs) as insufficient and therefore felt precarious about exercising because of this, as well as because previously, exercise during pregnancy was deemed a ‘taboo subject’. These uncertainties led to some participants avoiding exercise due to fear of miscarriage or falling in pregnancy.


*“You do get scared to move, whereas it’s just having that person to say, “Actually, no, it’s fine. You can go for a swim, you can do everything”, or to say, “No, don’t”, but it would have been nice to get advice on what we could do.”*
(Participant 3)

ii.Credibility: difficulty trusting the information received

In this subtheme, participants reflected on the different sources of information they received antenatal exercise advice from. Many participants expressed difficulty trusting the sometimes contradictory information they found from various sources.


*“I think, understandably, there’s a real mixed bag of guidance, and it’s not clear.”*
(Participant 11)

Credibility of the information pregnant females received regarding PA was a key influence for participants to perform antenatal exercise. Participants referred to feeling reassured receiving advice from HCPs or pregnancy exercise specialists, whilst feeling discouraged by ill-informed advice they heard from friends or social media.


*“You know, for me, it just felt like I think it just made it put me off a little bit because I’d like to say I’d rather speak to someone about it than just put on some random American woman on YouTube who’s telling me that this is fine.”*
(Participant 2)

#### 3.2.3. Theme 3: Motivators to Antenatal Exercise

This theme reflected on participant’s motivations to perform antenatal exercise and the reasons behind their activity choices. This included performing exercise to help make labour and the return to exercise at postpartum easier. It also included understanding of time and resources for antenatal exercise and how this shaped their accessibility to exercise or not. For instance, many pregnant participants recognised lack of time as a barrier to performing antenatal exercise due to occupation and childcare, though others identified antenatal exercise as a source of enjoyment and an important time to focus on themselves.

i.Training for the future self

In this subtheme, participants expressed their intention to exercise during pregnancy to make labour easier, due to recognition of labour as a feat of physical endurance. Participants expressed viewing pregnancy as a significant time to prepare themselves physically for birth, consequently physical fitness and therefore antenatal exercise were deemed an important part of many participant’s pregnancies.


*“This is one of the reasons that we do our walks and I make sure that we get out a lot, because labour is so intense, and physically one of the hardest things you can do, and I think it feels like the better trained you are generally and the fitter you are generally, the easier the labour will be.”*
(Participant 5)

Participant’s views on the importance of exercise to maintain physical fitness was not only to help make labour easier but also to help manage health in the postpartum period. Participants expressed their intention to exercise during pregnancy to make the transition back to being active and return to their pre-pregnancy body mass after labour easier, therefore viewing pregnancy as an important time to prepare for postpartum and maximise future benefits.


*“What I’m also conscious of is, I want to be able to pick my exercise back up after I’ve had the baby. So I don’t want there to be a huge gap where I’ve done nothing, so I figured it would be easier if I can just keep going ticking over as much as I can, then hopefully I’ll be able to get back into it a bit more easily after the baby’s here.”*
(Participant 12)

Participants frequently reported the importance of exercise for weight management pre-pregnancy. Some participants therefore ceased exercising to the same extent when COVID-19 lockdown measures were in place, as they would not be seeing anyone or going anywhere. Furthermore, a focus on weight management caused some participants to not perform antenatal exercise as they reported feeling a lack of control regarding their weight during this period and therefore lacked motivation.


*“Obviously during COVID, I couldn’t really do much. I started doing some things at home, and I was still hopeful that we could go on holiday in the summer, so I was doing the online classes, but then I think it got to about June, and then I thought, “Well, I’m definitely not going on holiday, so what’s the point?” So I stopped.”*
(Participant 9)

ii.Having time and resources to exercise

In this subtheme, many participants acknowledged that accessibility had a significant influence on their PA during pregnancy, whether that be access to time or facilities. Having access to a gym or classes was perceived as a helpful facilitator to opportunities to exercise. Yet, lack of time, bad weather and feeling that safety was compromised were viewed as barriers to accessing these opportunities.


*“So the main barrier is probably by the time the kids are in bed, it’s too dark to actually go out and enjoy the rest of the evening safely.”*
(Participant 8)

Due to the COVID-19 pandemic, many facilities were closed and participants in some instances felt uncomfortable exercising in public places with many people or vehicles close by, which restricted participant’s ability to perform antenatal exercise. Difficulty finding time to exercise was exacerbated for some participants during the COVID-19 pandemic because of home-schooling responsibilities and work commitments. However, other participants felt that they had more time due to the pandemic as they were able to work flexibly at home with significantly fewer social engagements to attend.


*“I’m lucky that I did live so close to the beach, so you could still go down the beach, but it did mean that there were more people going down to those places, so I try and avoid that on nice days. So if it’s really sunny or anything, I wouldn’t go, just because there’s too many people.”*
(Participant 13)

iii.Having someone there

This subtheme was identified as participants expressed feeling that support received from family and friends facilitated antenatal exercise. Some participants reported that sports clubs and competitions were useful to facilitate PA and were deemed a supportive network and encouraged them to exercise.


*“Yes, I think if you’re doing it by yourself, it’s easy to fall off the wagon, whereas if you’ve got that second person to kind of like, “Come on, let’s go, let’s do this”, or, “Let’s make sure we’re whatever”, then there is that extra motivation to carry on. So yes, I think that really makes a difference.”*
(Participant 8)

Yet, attending sports clubs and competitions was, at points, prohibited due to COVID-19 measures. Moreover, some participants stated they remained uncomfortable returning to sports clubs once restrictions lifted due to fear of contracting COVID-19.


*“I just felt mixing with thirty people just, I just didn’t fancy it, to be honest, so I’ve not played hockey or done any sort of team sports or group activities for the whole of the Coronavirus pandemic”*
(Participant 11)

Additionally, some participants noted that support from HCPs provided encouragement for them to perform antenatal exercise. Yet, other females, such as friends, acquaintances, and colleagues, expressing opposition to antenatal exercise rendered feelings of uncertainty amongst females.


*“When I’m running and people do say to you, you shouldn’t be doing that. It’s not very often to be honest, most people are very supportive, but it can put doubt in your mind if you’re doing the right thing because the last thing you want to do is put your baby at risk as a mum of course, you know, but am I being reckless, you know, should I be wrapped up in cotton wool eating lots of chocolate or whatever.”*
(Participant 14)

Despite not always wanting to be active during pregnancy, many participants felt it was important to do some activity rather than none as others were relying on them. This included ensuring pets were walked and adequately cared for as well as ensuring other children were getting fresh air and sufficient activity throughout lockdowns and social restrictions. This also included participants wanting to be active due to the viewed importance for a healthy pregnancy.


*“Yes, I think it’s just your wellbeing. It’s an improvement on your general wellbeing, and getting the whole family out as well for walks. It’s good for the whole household.”*
(Participant 10)

#### 3.2.4. Theme 4: A Process of Adaptation and Adjustment

Adaptation to pregnancy influenced participant’s ability to perform antenatal exercise. Many participants recognised the requirement to adjust their activity levels during pregnancy, though adjusting to pregnancy itself and symptoms they then experienced throughout pregnancy presented a ‘mental battle’ in some instances.

i.Adjusting to becoming pregnant

In this subtheme, some participants acknowledged that adjusting to becoming pregnant took time and significantly affected their daily ability more than they’d initially expected.


*“I’m just like a little bit pissed off with myself, just disheartened with how little I can actually do”*
(Participant 4)

Therefore, adapting during pregnancy in some instances presented obstacles in adjusting activity levels.


*“I’m getting bigger by the day. I think that’s going to, because of the type of exercise I do, that’s a bit of a challenge. Walking, I’m sure it’ll be fine. Running, I think is going to get more difficult”*
(Participant 12)

ii.Adapting activity whilst experiencing pregnancy symptoms

It was not only adjusting to pregnancy itself but navigating and adapting in the context of pregnancy symptoms. A number of participants reported struggling to exercise during pregnancy due to experiencing pregnancy symptoms including nausea, pelvic girdle pain and fatigue. These symptoms typically caused participants to adjust their activity (lowering intensity and duration), though sometimes resulting in a cessation of antenatal exercise.


*“I couldn’t really exercise in the mornings because I felt really ill, so I’d changed to going and doing stuff at lunchtime.”*
(Participant 7)

There were some pregnant participants, contrarily, who stated that exercise helped to relieve these physical symptoms.


*“Even if I do get ten minutes, and then I’ll feel better for doing it, but it’s just finding the energy to do it in the first place, but there’s definitely lots of benefits for it.”*
(Participant 6)

## 4. Discussion

This study explored the perceptions of antenatal exercise in pregnant females and the impact of the COVID-19 pandemic. Data suggests that to encourage pregnant females to perform antenatal exercise, access to credible information and better education is essential. Support and reassurance for pregnant females from friends, family, and HCPs, was important to reduce distress during pregnancy. However, access to credible information and support from HCPs was limited during the COVID-19 pandemic and therefore exacerbated barriers to antenatal exercise. Findings suggest that interventions exploring protective strategies, such as peer support and education, to improve wellbeing in this population, are warranted.

Understanding pregnant female’s perceptions of antenatal exercise is paramount to improve maternal and infant health. Previous studies, using a combination of qualitative data and data derived from questionnaires, have reported the intensity of perceived barriers to antenatal PA differ according to pre-pregnancy activity levels and have demonstrated that previous PA is one of the strongest predictors for maintaining activity during pregnancy [8,26,27]. For example, a randomised controlled trial in 399 Finnish antenatal females revealed that pregnant females who meet the PA recommendations prior to pregnancy maintained and increased their PA during the antenatal period [27]. Our data collected during the COVID-19 pandemic align with previous findings and suggest female’s identity of being an active or inactive person typically shaped their exercise intentions and behaviours during pregnancy. Moreover, previous activity habits prior to pregnancy appeared to determine the perceived importance of exercise and recognition of associated benefits. Therefore, promotion of a physically active lifestyle in females planning to conceive could be an effective strategy to increase antenatal exercise. Promotion materials to increase antenatal exercise levels should be framed to align with these findings and specifically target those who are not active already.

Antenatal HCPs, namely midwives, are key sources of information and guidance for pregnant females in the UK. It has been suggested that midwives’ knowledge of the National Institute for Health and Care Excellence (NICE) PA in pregnancy guidelines is limited [28]. Recent research investigating the promotion of PA by midwives to pregnant females identified a limitation in training, knowledge, confidence, time, resources, and perceptions of vulnerability as barriers to effective PA promotion [29]. The current study has identified the importance of validation to exercise from HCPs to encourage antenatal exercise participation. Further, participants perceived the advice received (or lack thereof) from HCPs about antenatal exercise as insufficient, which left them feeling precarious about exercising. Many participants reported turning to internet search engines, speaking to others, observing what other pregnant females do and popular culture books for antenatal exercise advice. Yet, participants expressed difficulty trusting the sometimes-contradictory information. The uncertainty amongst some pregnant participants led to exercise avoidance altogether. Participants felt that receiving advice from HCPs or pregnancy exercise specialists was reassuring and validating. However, the COVID-19 pandemic meant that opportunities for HCPs to encourage prenatal exercise were limited further due to fewer, shorter, and remote appointments. Regardless of the pandemic, efforts to enhance pregnant female’s awareness and understanding of the evidence-backed PA guidelines should be considered a priority, central to this is HCP and midwifery education on antenatal PA and exercise.

To understand why some pregnant females exercise whilst others do not, it is important to explore motivators, barriers, and facilitators to antenatal exercise. Most studies reported decreases in PA and increases in sedentary behaviours during COVID-19 lockdowns in several populations [30]. Weight management and body image have previously been reported as important motivators to exercise in non-pregnant [31] and pregnant [9] females. In the current study, some participants reported ceasing exercising when COVID-19 lockdown measures hit due to not seeing anyone and therefore losing motivation to exercise for weight management. Furthermore, the inability to control weight during pregnancy led to some participants losing exercise motivation. Working from home was also perceived as a barrier to PA during the COVID-19 pandemic for some participants as PA ordinarily accumulated during commuting decreased, whilst for others, working from home allowed flexibility for fitting in exercise in the working day. Moreover, accessibility to time and resources are important influencers for PA behaviour during pregnancy [9]. Closures of facilities, sports clubs and schools restricted several participant’s abilities to exercise. Closures of gym and exercise facilities required individuals to exercise in public or at home; this presented barriers for those participants without sufficient space to exercise at home or those who felt uncomfortable exercising in public. School closures caused those with other children to bear home-schooling responsibilities, which limited their time to exercise. Though, for some participants, ensuring the children were getting fresh air and sufficient activity throughout lockdowns and restrictions was a motivator for exercise. Therefore, the COVID-19 pandemic presented multiple barriers, as well as motivators, to antenatal exercise. Given the known benefits of antenatal exercise, it is recommended that antenatal HCPs promote ways of increasing PA with barriers associated with the COVID-19 pandemic in mind.

A recent systematic review identifying predictors of physical inactivity among pregnant females explains that alongside pre-pregnancy exercise status, sociodemographic, socioeconomic, lifestyle and health-related factors all contribute to activity levels during pregnancy [32]. As reflected in the current study, Yusof and colleagues identified that pregnancy symptoms such as nausea, fatigue and pelvic girdle pain can prevent pregnant females from being active [32]. Parity is another important factor that predicts PA during pregnancy [32]. Previous research has found that a parity ≥ 1 is inversely associated with regular antenatal exercise [33,34]. Similarly, a BMI indicating underweight, overweight, or obese has been inversely associated with regular antenatal exercise [33,34]. In the current study, both BMI and parity did not appear to have an impact on antenatal exercise levels recorded via the IPAQ. Though, interviews revealed that participants who already have children at home felt that finding time to exercise due to extra childcare commitments as a result of COVID-19 measures was a major limiting factor for antenatal exercise. Importantly, however, the current study did not measure pre-pregnancy exercise levels. Due to the possibility of further COVID-19-related restrictions or another similar pandemic, the promotion of mobile health technologies, such as smartphone applications or online fitness videos, that are tailored to pregnant females are recommended. Technology-based interventions aiming to increase antenatal exercise levels should consider pregnancy symptoms, parity, and BMI.

### Strengths and Limitations

In-depth, rich data was obtained from the interviews conducted. Voices of both primiparous and multiparous females with varying BMI, IMD and IPAQ scores were heard. The use of remote interviewing allowed nationwide participation from females who were experiencing varying levels of restrictions, therefore allowing us to capture a wide range of pregnancy experiences across the UK. It is noteworthy that most participants recorded being White British, highly educated and either married or in a relationship and living together. Therefore, views of pregnant females from other backgrounds are not incorporated in these data. Moreover, the larger observational trial that the current study was a part of, examined the impact of PA on glucose control throughout pregnancy. Due to the focus on PA, it is likely that participants who volunteered for the study were interested in PA. Nevertheless, the current study sample for interviews is not dissimilar to a lot of existing samples of pregnant females in this area. Additionally, potential external influence from other individuals in participant’s vicinity during the remote interviews may have restricted participants to feel that they could openly share feelings regarding peer support. Member checking provides participants with the opportunity to engage with, and add to, their own interview data. This was not performed in the current study due to challenges with ethical issues as a number of questions within the interview schedule addressed feelings towards COVID-19 lockdowns and the healthcare experience during COVID-19 (unpublished study). These questions prompted emotional responses; therefore, member checking was not performed to avoid reminding participants of previous distress.

## 5. Conclusions

The findings of this study highlight the importance of direct psychosocial support and clear, trustworthy information for pregnant females. Additionally, this study highlights the exacerbated barriers to antenatal exercise apparent during the COVID-19 pandemic, including reduced accessibility, time, and motivation. Findings can be used to inform development of comprehensive and appropriate promotional strategies for antenatal exercise. Future research should aim to design and systematically evaluate the implementation and impact of such strategies that take into consideration pregnant female’s perceived barriers and facilitators to exercise.

## Figures and Tables

**Table 1 ijerph-19-10635-t001:** Participant Demographic Information.

Characteristics	*n* (%)
Age (years)	
30–34	*n =* 8 (57%)
35–38	*n =* 6 (43%)
Week Gestation	
13–20	*n =* 3 (21%)
21–28	*n =* 5 (36%)
29–37	*n =* 6 (43%)
Pre-pregnancy BMI (kg/m^2^)	
Healthy Weight (18.5–24.9 kg/m^2^)	*n =* 8 (57%)
Overweight (25–29.9 kg/m^2^)	*n =* 4 (29%)
Obese (>30 kg/m^2^)	*n =* 2 (14%)
Highest Level of Education	
GCSEs or Equivalent	*n =* 1 (7%)
Undergraduate Degree	*n =* 6 (44%)
Postgraduate Degree	*n =* 2 (14%)
Doctorate or Equivalent	*n =* 3 (21%)
Other/Prefer not to say	*n =* 2 (14%)
Parity	
0	*n =* 7 (50%)
1	*n =* 6 (43%)
2	*n =* 1 (7%)
IMD Quintile	
1	*n =* 3 (21%)
2	*n =* 4 (30%)
3	*n =* 3 (21%)
4	*n =* 2 (14%)
5	*n =* 2 (14%)
IPAQ Category	
Low	*n =* 5 (36%)
Moderate	*n =* 8 (57%)
High	*n =* 1 (7%)

Abbreviations: IPAQ, International Physical Activity Questionnaire; BMI, Body Mass Index; IMD, Index of Multiple Deprivation taken from Office of National Statistic Indices of Multiple Deprivation (2010) [25].

**Table 2 ijerph-19-10635-t002:** Main themes and sub-themes.

What Are Pregnant Female’s Perceptions of Antenatal Exercise?
1. Perceptions of being an ‘active person’ shaping activity levels in pregnancy Previous activity habits shaping current habits Seeing the value in being active: the perceived importance of exercise
2. How do I know what is right? Uncertainty, seeking validation and feeling informed. Uncertainty and seeking validation Credibility: difficulty trusting the information received
3. Motivators to antenatal exercise Training for future self Having the time and resources to exercise Having someone there
4. A process of adaptation and adjustment Adjusting to becoming pregnant Adapting activity whilst experiencing pregnancy symptoms

## Data Availability

The data presented in this study are available on request from the corresponding author. The data are not publicly available due to sensitive topics discussed by participants.

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
