# Peer review of "Perceptions of Antenatal Exercise in Pregnant Females and the Impact of COVID-19"

_ijerph, 2022, doi:10.3390/ijerph191710635_

Round 1

Reviewer 1 Report

The idea for the study is interesting, however, the study group is so small that it can only be called "a preliminary study". The results are presented more like a story than scientific data - the authors are constantly quoting the respondents, which should not happen. There are no comparisons - but they probably could not have been done because of 14 participants only. Only some parts of the interviews referred to covid-19 pandemic, so the title is also questionable. 

Author Response

The authors thank the reviewer for their time spent reading the manuscript and assessing the methods and results.

Within reflexive thematic analysis of qualitative research an in-situ decision regarding the final sample size is shaped by the adequacy (richness, complexity) of the data for addressing the research question (Braun & Clarke, 2019). Moreover, saturation is the most common guiding principle for assessing sample size in qualitative research, which refers to the point in data collection when no additional issues or insights are identified. A recent systematic review of empirical tests showed saturation was reached in 9-17 interviews amongst studies (Hennink & Kaiser, 2022). In the current study, uniform perspectives and consistent discussions were reported, therefore saturation was defined at 14 participants.  

The authors feel that quotes are necessary in the current study to demonstrate the audit trail and illustrate themes, as is standard practice in qualitative research publications. Braun and Clarke (2012) state in their Thematic Analysis Handbook that “each extract would ideally provide a vivid, compelling example that clearly illustrates the analytic points you are making”. Moreover, as part of the analytic procedure, disconfirmatory evidence was reported throughout the text (we have listed some examples below to demonstrate this in the context of our manuscript).

Line 260-263: For instance, many pregnant participants recognised lack of time as a barrier to performing antenatal exercise due to occupation and childcare, though others identified antenatal exercise as a source of enjoyment and an important time to focus on themselves. 

Line 305-309: Difficulty finding time to exercise was exacerbated for some participants during the COVID-19 pandemic because of home-schooling responsibilities and work commitments. However, other participants felt that they had more time due to the pandemic as they were able to work flexibly at home with significantly fewer social engagements to attend.

Line 433-436: Working from home was also perceived as a barrier to PA during the COVID-19 pandemic for some participants as PA ordinarily accumulated during commuting decreased, whilst for others, working from home allowed flexibility for fitting in exercise in the working day.

Line 449-458: Parity is another important factor that predicts PA during pregnancy (Yusof et al., 2020; Chasan-Taber et al., 2007). Previous research has found that a parity ≥1 is inversely associated with regular antenatal exercise (Owe et al., 2009; Juhl et al., 2012). Similarly, a BMI indicating underweight, overweight, or obese has been inversely associated with regular antenatal exercise (Owe et al., 2009; Juhl et al., 2012). In the current study, both BMI and parity did not appear to have an impact on antenatal exercise levels recorded via the IPAQ. Though, interviews revealed that participants who already have children at home felt that finding time to exercise due to extra childcare commitments as a result of COVID-19 measures was a major limiting factor for antenatal exercise. Importantly, however, the current study did not measure pre-pregnancy exercise levels.

Many data, however, were in parallel. To ensure disconfirmatory evidence is reported within the manuscript, we have added the text ‘Disconfirmatory evidence, where identified, is highlighted within the text’ on line 133-144.

Lastly, as the interviews were performed during the COVID-19 pandemic where many national COVID-19 restrictions were in place (November 2020-May 2021), the authors feel it is of paramount importance that COVID-19 is a main theme of the study to acknowledge the context in which the study was performed. The pandemic had significant consequences for many pregnant females such as changes in childcare responsibilities, working from home, and closures of facilities. Therefore, without acknowledging COVID-19 within the title and throughout the study, the authors feel the findings would be misleading.

Reviewer 2 Report

This was a well-written manuscript that provides important findings in relation to antenatal exercise during pregnancy and the impact of the pandemic.

Consider replacing wording "face-to-face" to "in-person".  Face-to-face can technically be via Zoom - you are looking at each others faces.

Please be more explicit of how you operationalised "purposive sampling" - as this may bias the results

Referencing formatting issues on line 80, 93, 97, 102, 124, 127

Remove "e.g." line 103

Please explain eligibility requirements (Lines 82 to 87).  

Member checking and lack thereof should be included in the limitations section of the paper.

Author Response

This was a well-written manuscript that provides important findings in relation to antenatal exercise during pregnancy and the impact of the pandemic. The authors thank the reviewer for clear and important feedback on the manuscript. We feel these changes will improve the quality of our manuscript.

Consider replacing wording "face-to-face" to "in-person".  Face-to-face can technically be via Zoom - you are looking at each other’s faces. Thank you for raising this. Admittedly, this is not something we had thought about but makes absolute sense; we have amended the use of language throughout the manuscript accordingly.

Please be more explicit of how you operationalised "purposive sampling" - as this may bias the results.  We have added further explanation to ensure this is explained sufficiently. Line 77-81 - Purposive sampling was employed by asking participants who were in various stages of gestation whether they would like to take part in the semi-structured interviews (i.e. if there were many participants in trimester 2 who had been interviewed, no more participants in trimester 2 were asked) to increase diversity/representation and enable sharing of broader experiences across the sample18.

Referencing formatting issues on line 80, 93, 97, 102, 124, 127 Thank you, we have addressed these issues.

Remove "e.g." line 103 Completed.

Please explain eligibility requirements (Lines 82 to 87).  Line 85-87 - These eligibility requirements were deemed appropriate as part of the (aforementioned) larger observational trial to enable assessment of relationship between PA and glucose control.

Member checking and lack thereof should be included in the limitations section of the paper. This has been acknowledged. Lines 483-489 - Member checking provides participants with the opportunity to engage with, and add to, their own interview data. This was not performed in the current study due to challenges with ethical issues as several questions within the interview schedule addressed feelings towards COVID-19 lockdowns and the healthcare experience during COVID-19 (unpublished study). These questions prompted emotional responses; therefore, member checking was not performed to avoid reminding participants of previous distress.

Reviewer 3 Report

The revised article aims to analyse and understand the views and perceptions of pregnant women's perceived barriers and limitations to prenatal physical activity and how these have been affected by the COVID-19 pandemic. This is a topic of interest given the beneficial relationship of exercise to the health of both mother and newborn, a topic that despite its importance is little addressed from the point of view of the perceptions of pregnant women themselves. They add a further point of interest by considering how the covid19 pandemic may have affected them, especially during times of confinement and restraint.

The article is clear, well-structured and methodologically well laid out. As qualitative research, the small number of participants in the study, a total of 14 pregnant women, is understandable, yet the experimental design is appropriate for the intended purpose.  The method of recruitment, data collection through semi-structured interviews and data analysis (reflexive thematic analysis) using Nvivo version 1.5 software are clearly specified. The aspects highlighted in the methodology undoubtedly allow the results to be reproduced without any problem, since the questions included in the interview are also provided.

The data provided in the tables are sufficient for qualitative research. They are easy to understand and interpret, although the analysis carried out is somewhat simple and presents little depth in the treatment of the thematic lines addressed in the survey.

The conclusions are appropriate to the results and discussion presented. And the limitations of the study and possible improvements are clearly presented.

The section that is undoubtedly the most limited is the bibliographical references, given that 43% of them correspond to years equal to or less than 2010, 23% to the period between 2011-2016 and only approximately one third (33%) correspond to the period between the last 5 years (2017-2022). It would be advisable to carry out a more updated review of the subject matter addressed.

Based on the above, it is recommended: Accept after minor revisions: try to update the bibliography.

Author Response

Thank you for your thorough review that will strengthen our manuscript. The bibliography has now been updated as requested. 31% correspond to years equal to or less than 2010, 23% to the period between 2011-2016 and 46% correspond to the period between the last 5 years (2017-2022).